# Treatment, Persistent Symptoms, and Depression in People Infected with COVID-19 in Bangladesh

**DOI:** 10.3390/ijerph18041453

**Published:** 2021-02-05

**Authors:** Md. Saiful Islam, Most. Zannatul Ferdous, Ummay Soumayia Islam, Abu Syed Md. Mosaddek, Marc N. Potenza, Shahina Pardhan

**Affiliations:** 1Department of Public Health and Informatics, Jahangirnagar University, Savar, Dhaka 1342, Bangladesh; m.zannatul.ferdous@juniv.edu (M.Z.F.); sumaiyaaislam@gmail.com (U.S.I.); 2Youth Research Association, Savar, Dhaka 1342, Bangladesh; 3Department of Pharmacology, Uttara Adhunik Medical College, Uttara, Dhaka 1230, Bangladesh; 4Quest Bangladesh Biomedical Research Center, Lalmatia, Dhaka 1207, Bangladesh; 5Department of Psychiatry, and Child Study Center, Yale School of Medicine, New Haven, CT 06510, USA; marc.potenza@yale.edu; 6Connecticut Mental Health Center, New Haven, CT 06519, USA; 7Connecticut Council on Problem Gambling, Wethersfield, CT 06519, USA; 8Department of Neuroscience, Yale University, New Haven, CT 06510, USA; 9Vision and Eye Research Institute, School of Medicine, Anglia Ruskin University, Young Street, Cambridge CB1 2LZ, UK; shahina.pardhan@aru.ac.uk

**Keywords:** depression, COVID-19, severe acute respiratory syndrome coronavirus 2, recovery, Bangladesh, sleep–wake disorders

## Abstract

*Background:* Coronavirus disease (COVID-19) has affected people’s lives globally. While important research has been conducted, much remains to be known. In Bangladesh, initial treatment (self-administered, hospitalized), persistent COVID-19 symptoms (“long COVID-19”), and whether COVID-19 leads to changes in mental state, such as depressive symptoms, of people are not known. This study aimed to examine treatment, persistent symptoms, and depression in people who had been infected with COVID-19 in Bangladesh. *Methods:* A cross-sectional survey was conducted on 1002 individuals infected with COVID-19 (60% male; mean age = 34.7 ± 13.9; age range = 18–81 years), with data taken over a one-month period (11 September 2020 to 13 October 2020). A self-reported online questionnaire was used to collect data on socio-demographics, lifestyle, COVID-19 symptoms (during and beyond COVID-19), medication (over-the-counter or doctor-prescribed), and depression (assessed using the Patient Health Questionnaire (PHQ-9)). *Results:* Twenty-four percent of participants self-medicated with over-the-counter medicine when they were first diagnosed with COVID-19. Self-medication was higher among female vs. male respondents (29.6% vs. 20.2%, respectively, *p* = 0.002). A minority (20%) reported that they experienced persistent COVID-like symptoms after recovering from COVID-19. The most reported persistent symptoms were diarrhea (12.7%) and fatigue (11.5%). Forty-eight percent of participants were categorized as having moderate to severe depression. Based on multivariate regression analysis, depression during COVID-19 was positively associated with lower family income, poor health status, sleep disturbance, lack of physical activity, hypertension, asthma/respiratory problems, fear of COVID-19 re-infection, and persistent COVID-19 symptoms. *Conclusions:* The findings suggest a need for appropriate interventions for COVID-19 patients to promote physical and mental wellbeing.

## 1. Introduction

After emerging in December 2019, the novel coronavirus SARS-CoV-2 (severe acute respiratory syndrome coronavirus 2) has had a significant impact on people globally, and the number of coronavirus disease (COVID-19) cases continues to increase. Almost all countries in the world have been affected by the coronavirus, with 98.85 million confirmed cases, including 2.12 million deaths, as of 13 January 2021 [1]. At the initial stage, most cases were suspected of having emerged from zoonotic exposure in Wuhan, China, and the virus has since been frequently transmitted from human to human, as it has a higher reproductive capacity than many other infectious diseases [2,3].

According to Bangladesh’s Institute of Epidemiology Disease Control and Research, the country confirmed its first case on 8 March 2020 [4,5,6]. Like other countries, Bangladesh adopted non-therapeutic measures to mitigate COVID-19 spread, such as maintaining spatial distancing, closing non-essential institutions, and not permitting mass gatherings [7]. Despite this, as of 23 January 2021, Bangladesh held the twenty-seventh position globally, with 531,326 confirmed cases, including 8003 deaths [8].

Over the last few months, millions of people globally have been hospitalized due to COVID-19 with multiple symptoms, including fever, dry cough, fatigue, diarrhea, dyspnea, headache, nausea, and vomiting [9]. Recovery is often not total. In Italy, only 13% of people were completely free of COVID-19 symptoms after 60 days, with 55% of people experiencing at least three persistent symptoms and 32% of people experiencing one or two persistent symptoms [10]. In the USA, COVID-19 patients with persistent symptoms reported experiencing a lower quality of life after two or three weeks [11]. Following the outbreak of the previous SARS virus, a chronic post-SARS syndrome was reported one year after infection [12], affecting people both physically and mentally [13]. Other studies of the SARS epidemic reported psychiatric concerns after the early recovery phase, reflected in increased anxiety and depression [14,15].

Self-medication is commonly performed globally. Over-the-counter medicines without a doctor’s prescription may negatively affect healthcare resources and patients’ health [16,17,18]. Factors that have been shown to influence self-medication include previous experience with a disease or a symptom, availability of medicine, increasing costs of prescription drugs, and the willingness of people to empower themselves [19,20]. In Bangladesh, self-medication is influenced largely by socioeconomic status, educational attainment, and lack of proper knowledge [21]. Although the World Health Organization (WHO) advocates self-medication for inhibiting and treating minor health problems at minimal cost, in severe cases, it may lead to adverse health outcomes, including higher mortality [22]. The current doctor-to-patient ratio in Bangladesh is 5.26 to 10,000, placing the country in the second-lowest position among South Asian countries [23]. Therefore, lack of knowledge, scarcity of health service resources, new treatment protocols, and unreliable sources of national or social media news may influence people’s decisions to self-medicate to treat symptoms of COVID-19, such as fever, sore throat, or dry cough. 

Pandemic issues such as spatial distancing, isolation, quarantine, and economic hardships may lead to or exacerbate depression, frustration, fear, grief, anger, shame, desperation, boredom, stress, and panic [6,24,25,26]. Depression is a relatively common mental health problem that many individuals will likely experience during and after the crisis [24]. Depression is a significant psychiatric illness affecting more than 264 million people around the world [27]. Depression affects individuals across the lifespan [28]. Depression is characterized by sadness that may include tearfulness, loss of interest or pleasure, guilt or low self-worth, trouble sleeping, decreased or increased appetite, low energy, diminished sex drive, and poor concentration [29]. In Bangladesh, depression has been observed in different populations during the COVID-19 pandemic, including the general public (47.2%) [30], university students (62.9%) [26], medical students (49.9%) [31], and healthcare workers (39.5%) [25]. Previous studies have reported several factors that are associated with depression, including female sex, living in an urban area, reduced or no physical exercise, dissatisfaction with sleep, self-reported moderate/poor health status, smoking, the presence of COVID-19 symptoms, and fear of COVID-19 re-infection [25,26,30,32,33]. A recent study has shown that 56.6% of 153 surveyed COVID-19 patients in Bangladesh reported depression [34].

To date, no prior study has investigated self-medication, hospitalization, persistent or ongoing symptoms, and ensuing mental wellbeing in people who were infected with COVID-19 in Bangladesh. The correlates of depression and how people respond to COVID-19 are important to understand given the potential impact of the viral illness on mood and potential relationships between how people respond to the infection or recover from it and their health and well-being. Such information may help policy, prevention, and treatment efforts. Thus, this study aimed to explore the characteristics of people in Bangladesh who were infected by COVID-19. We were interested in understanding factors relating to depression, self-medication, hospitalization, and persistence of COVID-19 symptoms.

## 2. Materials and Methods

### 2.1. Participants and Procedure

A cross-sectional survey was conducted over a one-month period (from 11 September to 13 October 2020) involving people who had previously tested positive for COVID-19 in Bangladesh. A pre-tested semi-structured questionnaire adapted from the literature [10] was used to collect data using the Google survey tool. The questionnaire was translated using the guidelines of back translation (e.g., Beaton et al. (2000)) [35], as previously used in Bangladesh [25,36,37]. The questionnaire was translated into Bangla (participants’ first language) and then translated back to English by experts with very good command of both the Bangla and English languages. A pilot test was conducted on 15 individuals to assess the questionnaire in terms of acceptability and clarity. Minor modifications were incorporated after the pilot. Individuals who had recovered from COVID-19 were then invited to complete the survey. Word-of-mouth and snowballing techniques were used to recruit participants. Initially, nearly 100 research assistants (RAs) were recruited from websites and social media through advertisements. Then, they (RAs) were properly instructed to share our survey link among their family members, friends, colleagues, and other peers who had recovered from COVID-19. This technique has been used in previous research in Bangladesh [31]. Approximately 1200 individuals were invited to participate in the survey, of whom 1083 individuals took part. After removing incomplete surveys, 1002 participants were included in the final sample. 

The inclusion criteria of the participants were as follows: ≥18 years old, having tested positive for COVID-19, and willingness to complete the survey. Exclusion criteria included having incomplete responses, responses from those who were not infected with COVID-19, and those who did not consent. 

### 2.2. Sampling Method

The sample size was calculated using the following equation:*n* = (*z*^2^pq)/*d*^2^; *n* = {1.96^2^ × 0.5 × (1 − 0.5)}/0.05^2^ = 384.16 ≈ 384(1)
where *n* = number of samples; *z* = 1.96 (95% confidence level); *p* = prevalence estimate (0.5); *q* = (1 − *p*); *d* = precision limit or proportion of sampling error (0.05).

As there had been no prior similar study focusing on individuals after recovery from acute COVID-19, we made a best estimate that 50% of the subjects would have psychological problems. Assuming a 10% non-response rate, a sample size of 423.5 ≈ 424 participants was estimated. Our sample size exceeded this estimate. 

### 2.3. Ethics

All procedures were conducted in accordance with the principle for human investigations (i.e., Helsinki Declaration). Formal ethical approval was granted by the Ethical Review Committee, Uttara Adhunik Medical College (Ref. no: UAMC/ERC/23/2020). Participants were informed about the procedure and purpose of the study and the confidentiality of the information provided. All participants consented willingly to participate. All data were collected anonymously and analyzed using a coding system. 

### 2.4. Measures

The self-report survey consisted of several sections: (i) socio-demographic and lifestyle information; (ii) health status, underlying health diseases, and COVID-19-related questions, including questions regarding persistent symptoms; (iii) treatment, including self-medication and hospitalization; (iv) a psychometric measure of depression using the Patient Health Questionnaire (PHQ-9) scale; and (v) an assessment of fear of COVID-19 re-infection. 

#### 2.4.1. Socio-Demographic and Lifestyle Information

Socio-demographic information obtained included age (later categorized into younger adults (18–39 years) and middle-aged/older adults (>40 years)), sex (male/female), height, weight, marital status (unmarried/married/divorced or widowed), family type (nuclear (two parents and their children)/joint (family unit with more than two parents, extended family)), monthly family income and residence (urban/rural). Socio-economic status (SES) was categorized into three classes: lower, middle, and upper, based on monthly family income (later categorized as lower (<15,000 Bangladeshi Taka (BDT) ≈ 177 US$); middle (15,000–30,000 BDT ≈ 177–354 US$); and upper (>30,000 BDT ≈ 354 US$)) [38,39]. Body mass index (BMI) was calculated as the ratio of weight in kilograms (kg) to the square of height in meters (m^2^). BMI was used to categorize participants as underweight (<18.5 kg/m^2^), lean (18.5–24.9 kg/m^2^), or overweight/obese (>25 kg/m^2^) [40,41,42].

Lifestyle-related questions assessing physical activity and smoking were asked with binary ‘yes/no’ responses. The average number of hours of sleep during the COVID-19 pandemic was obtained through three possible responses: normal (7–9 h), less than average (<7 h), or more than average (>9 h) [26,43]. For analysis, less than average (<7 h), and more than average (>9 h) sleep hours were classified as sleep disturbances.

#### 2.4.2. COVID-19 Symptoms and Initial Treatment Measures 

Self-reported health status was later categorized as good and moderate/poor (“moderate” and “not good”). Participants also used ‘yes/no’ questions to indicate whether they suffered from any underlying chronic diseases (diabetes, hypertension, heart problems, cancer, kidney disease, asthma/respiratory problems).

Data were also obtained using ‘yes/no’ responses to determine the type of COVID-like symptoms (e.g., fever, fatigue, cough, headache, lack of appetite, sore throat, myalgia, dyspnea, sputum production, chest pain, and/or diarrhea) both during and after the COVID-19 stage.

#### 2.4.3. Treatment, Including Self-Medication and Hospitalization

Participants were asked whether they were hospitalized due to COVID-19. Information on self-medication was ascertained using a question, “*Did you take any medication for COVID-19 symptoms*?”, with three possible answers: treatment from doctor, self-medication, and no action (rest). Reasons for self-medication were also ascertained.

#### 2.4.4. Patient Health Questionnaire (PHQ-9) Scale 

To ascertain the level of depressive symptoms, a validated Bangla version of the PHQ-9 [44] was used [45,46]. The PHQ-9 consists of nine-item questions on sleep, exhaustion, changes in appetite, difficulties with concentration, and suicidal thoughts over the past two weeks. An example question is “*Do you have t**rouble falling or staying asleep, or sleeping too much?*”, with a four-point Likert scale ranging from 0 (“*Not at all*”) to 3 (“*Nearly every day*”) [47]. The total score was calculated by summing the raw scores of each item ranging from 0–27, with a higher score indicating greater severity of depression. The levels of depressive symptoms were classified into five groups according to their score as minimal (0–4), mild (5–9), moderate (10–14), moderately severe (15–19), and severe (20–27). In the present study, the PHQ-9 scale was found to have Cronbach’s α of 0.86. 

#### 2.4.5. Fear of COVID-19 Re-Infection 

Participants were asked, using yes/no questions, whether they were worried about re-infection with COVID-19 (e.g., “*Do you fear that you might get infected with COVID-19 again?*”).

### 2.5. Statistical Analyses

Descriptive statistics (e.g., frequencies, percentages, means, standard deviations) and some first-order analyses (e.g., Chi-square tests, Fisher’s exact tests) were performed. Variables that significantly differed in bivariate analysis were included in multivariate regression analysis. Bonferroni-corrected *p*-values (*p* < 0.003) were used to determine statistical significance. All data were analyzed using two statistical software packages (Microsoft Excel 2019 (Microsoft Corporation, Albuquerque, NM, USA) and IBM SPSS Statistics version 25 (Armonk, NY, USA)).

## 3. Results

### 3.1. Participants

The present sample included 1002 participants with a mean age of 34.7 years (*SD* = 13.9), with age ranging from 18–81 years. Table 1 provides data from participants who reported a confirmed test for COVID-19.

Study participants were characterized by the following attributes: relatively younger age (<40 years) (65.2%), male (57.9%), married (53.1%), from nuclear families (71.0%), and lean (47.7%) and overweight/obese (47.3%) BMIs. More than half of the participants experienced sleep disturbances (52.2%), and most did not engage in physical exercise (72.2%) during the COVID-19 pandemic. Approximately one-eighth (12.3%) of participants smoked. Participants reported the following conditions: diabetes (21.1%), hypertension (24.9%), heart disease (8.2%), cancer (2.4%), kidney problems (5.9%), and asthma/respiratory problems (25.4%). 

### 3.2. Hospitalization Due to COVID-19 

Twenty-one percent of individuals reported having been hospitalized due to COVID-19. Bivariate analysis shows that that hospitalization was significantly higher among middle-aged/older, female and married individuals, those residing in joint family units and reporting self-medication, and those with lower SES, overweight/obesity, sleep disturbances, chronic underlying diseases (e.g., diabetes, heart disease, cancer, and kidney problems), fear of COVID-19 re-infection, higher depression scores, and persistent COVID-19 symptoms (Table 2). 

Adjusted logistic regression using only those variables that were significant in bivariate analysis retained middle-aged/older adult, lower SES, and persistent symptoms as factors associated with COVID-19 hospitalization (Table 3).

### 3.3. Self-Medication for COVID-19 

Participants’ initial treatment after identifying COVID-19 symptoms included consulting a doctor (67.7%), self-medicating (24.2%), and rest for relief (8.2%). Bivariate analyses showed the proportion of people who self-medicated was significantly higher among females, individuals of middle-age/older years and lower SES, those who reported hospitalization, poor health, underlying health conditions (e.g., diabetes, cancer, and kidney problems), moderate to severe depression, and persistent symptoms (Table 2). Adjusted multiple regression, using only variables that were significant in bivariate analysis, retained lower SES and persistent COVID-19 symptoms as variables significantly associated with self-medication (Table 3). 

Appendix A questions around reasons for self-medication or unwillingness to seek healthcare consultation included inadequate local medical services (52.4%), dissatisfaction with local healthcare services (37.4%), cost of consultation with doctors (12.5%), perception of illness as being non-threatening (32.9%), lack of time (16.8%), and lower urgency (24.3%).

### 3.4. Persistent Symptoms (Long COVID-19)

Twenty percent of participants reported that they experienced persistent and ongoing COVID-like symptoms. Figure 1 summarizes COVID-19 symptoms during the COVID-19 and post-COVID-19 stages. The most frequently reported symptoms during COVID-19 were fever (89.1%) and fatigue (80.8%). In contrast, participants reported diarrhea (12.7%) and fatigue (11.5%) as the most frequent persistent symptoms.

The proportion of persistent symptoms was higher among middle-aged/older and female individuals, those living in joint families, and those reporting lower SES, poor health status, overweight/obesity, sleep disturbances, chronic underlying diseases (e.g., diabetes and cancer), fear of COVID-19 re-infection, and moderate to severe depression; self-medication and hospitalization were also associated (Table 2). Adjusted multiple regression, using only variables that were significant in bivariate analyses, retained upper SES, fear of COVID-19 re-infection, hospitalization, and self-medication as variables significantly associated with persistent symptoms (Table 3).

### 3.5. Depressive Symptoms during COVID-19 

As per PHQ-9 designation, depression levels were observed as minimal (30.4%), mild (21.5%), moderate (24.2%), moderately severe (19.4%), and severe (4.6%). The prevalence estimate of moderate to severe depression was 48.2%. The proportion of moderate to severe depression was higher among female and married individuals, those of lower SES, with moderate/poor health, with sleep disturbance, not engaging in physical exercise, having underlying health conditions (e.g., diabetes, hypertension, heart disease, cancer, and asthma/respiratory problems), reporting self-medication, having persistent symptoms, with fear of COVID-19 re-infection, and who were hospitalized. Adjusted multiple regression, using only variables that were statistically significant in bivariate analysis, retained lower SES, poor health, sleep disturbances, asthma/respiratory problems, and fear of COVID-19 infection as factors significantly associated with moderate to severe depression (Table 1).

## 4. Discussion

In this study, we examined correlates of self-medication, hospitalization, persistent symptoms, and depression with COVID-19 among hospitalized and non-hospitalized individuals. More than 50% of respondents reported that their health was not good, with sleep disturbances and difficulties engaging in physical activity. Additionally, diabetes, hypertension, and asthma/respiratory problems were among the most frequent chronic diseases reported by participants.

Twenty-one percent of individuals reported having been hospitalized for COVID-19. Significant odds ratios (ORs) of hospitalization were associated with middle aged/older adults (OR = 2.97; 95% CI = 1.85–4.77), lower SES (OR = 2.04; 95% CI = 1.28–3.23), and persistent symptoms (OR = 4.9; 95% CI = 3.19–7.51). These findings resonate with those from a study from the Dhaka Medical College regarding COVID-19 [48].

Twenty-four percent of people self-medicated after identifying COVID-19 rather than seeking consultation during the pandemic. This figure is lower than previous studies conducted in Bangladesh among patients attending outpatient departments (69.0%) [49] and undergraduate students (88.0%) [50]. Since our study reflected a greater proportion of younger adults, the lower rate of self-medication might reflect this demographic variable. Other reasons may include fear of the disease and lack of familiarity with medicines. Hence, individuals may be more likely to consult with their doctor to avoid complications. Comparisons with other studies may be challenging as it is possible that the medication taken for COVID-19 might be in addition to others they may be taking, and we did not collect these data. Self-medication was significantly associated with lower SES (OR = 2.31; 95% CI = 1.56–3.41) and the presence of persistent symptoms (OR = 2.14; 95% CI = 1.44–3.18). Reasons for self-medication included limited access to healthcare facilities (52.4%), costs involved in consultation or hospital fees (12.5%), time constraints (16.8%), and lower perception of illness (32.9%). Similar reasons have been reported in several studies conducted in northern England and Australia [51,52] and in low- and middle-income countries [53,54]. Other studies conducted in Bangladesh, Ethiopia, and Poland suggest that self-medication practices were influenced by the lower cost of medicine, time restrictions, whether an illness was considered not to be serious, family size, and religious beliefs [50,55,56]. The proportion of people impacted by COVID-19 was particularly high in Bangladesh (particularly Dhaka) during this pandemic [57]. The inappropriate use of antimicrobials along with other supplementary medicines, without consulting a doctor, may increase the possibility of drug–drug interactions, hide symptoms of the COVID-19 disease, and promote the development of antimicrobial resistance [58,59,60]. In Bangladesh, there is evidence of increased resistance to antibiotics, as sales of medicines are not restricted [61].

In the present study, nearly two-thirds of participants (62.8%) reported fear of re-infection of COVID-19, and 20% experienced persistent symptoms of COVID-19. The most frequently reported persistent symptoms were diarrhea (12.7%) and fatigue (11.5%) during the post-COVID-19 stage. Factors associated with persistent symptoms were better SES (OR = 0.45; 95% CI = 0.28–0.72), fear of re-infection (OR = 3.32; 95% CI = 2.01–5.47), self-medication (OR = 2.23; 95% CI = 1.47–3.37), and hospitalization (OR = 4.89; 95% CI = 3.22–7.44). Persistent symptoms have also been reported in Italy, where 87.4% of people reported at least one symptom (e.g., asthma, fatigue) after recovery from COVID-19 [10]. A study from the Netherlands and Belgium also reported fatigue and asthma as the most common symptoms among non-hospitalized individuals three months after the onset of COVID-19 [62]. In those who were hospitalized due to COVID-19, persistent symptoms included fatigue, dyspnea, memory loss, poor sleep, and concentration difficulties [62]. It is therefore important that follow-up care be offered to individuals who have been ill with COVID-19 [63].

Our study revealed considerable depression (48%) among people who had been infected with COVID-19. It is possible that some individuals had a previous history of depression and therefore the data captured might not be due to COVID-19. However, our data agree with published data on depressive symptoms during COVID-19 in Bangladesh in the general population (47.2%) [30], healthcare workers (39.5%) [25], and medical students (49.9%) [31]. It may not be valid to compare pre-COVID rates of depression, as cohorts may be different and there could be significant variations in demographic factors such as age, sex, marital status, and other features. Our findings show positive associations between depression and lower SES (OR = 2.22; 95% CI = 1.51–3.26), good health status (OR = 3.17; 95% CI = 2.33–4.33), sleep disturbances (OR = 1.96; 95% CI = 1.46–2.64), asthma/respiratory problems (OR = 1.73; 95% CI = 1.23–2.43), and fear of COVID-19 re-infection (OR = 2.16; 95% CI = 1.58–2.97). Several previous studies in Bangladesh conducted during the COVID-19 pandemic reported that depression was associated with moderate/poor health status, sleep concerns, chronic diseases, smoking, and fear of COVID-19 re-infection [25,26,30,32,33,64]. Another study also suggested that lower family income (OR = 2.99, 95% CI = 1.11–8.05), persistent symptoms (OR = 2.63; 95% CI = 1.34–5.17), and poor social support (OR = 3.13; 95% CI = 1.34–7.30) were significantly associated with depression [65]. Negative emotions linked to COVID-19 may underlie associations with depression [66]. Our findings agree with data showing frequent depression, panic, and general anxiety disorder in Bangladesh during the COVID-19 pandemic [67,68]. A recent study suggested that the pandemic and economic recession would have a further negative impact on people who already suffer from mental illness and substance use disorders [69]. According to the WHO, mental health services have been disrupted by COVID-19 in almost 93% of countries globally [70], with 130 countries showing reduced access to services related to mental health [71]. The WHO also states that people with pre-existing mental, neurological, and substance use disorders have a high risk of mortality during the pandemic [70]. Some recent studies of the COVID-19 pandemic and psychological concerns in Bangladesh have reported high rates of depression among different populations [25,26,30,64].

### Limitations

Study limitations should be acknowledged when interpreting findings. First, the study was cross-sectional and thus cannot determine causality. The findings may not reflect the impact of COVID-19, especially for depression, which may have developed prior to the COVID-19 pandemic. Longitudinal studies are needed to determine sequential relationships. Second, this study used an online self-reporting approach that can be vulnerable to biases (e.g., social desirability, memory recall). However, compared to offline and/or face-to-face surveys, anonymous questionnaires may increase veracity of responses. Our study had a high proportion of younger adults and therefore future larger-scale studies involving a wider age range are warranted. Additionally, the extent to which the findings generalize to other populations outside of Bangladesh warrants further investigation using more sophisticated modeling as described elsewhere [72]. In addition, we did not determine the time frame for when respondents had been diagnosed with COVID-19 prior to data collection, and future studies should gather such data.

## 5. Conclusions

The findings of this study contribute to the literature in terms of initial treatment, persistent symptoms, and mental health concerns among individuals who were infected with COVID-19 in Bangladesh. The reports indicate that 12.7% of participants reported diarrhea (12.7%) and fatigue (11.5%) as the most frequent persistent symptoms (“long COVID-19”), and nearly half of participants experienced moderate to severe depression. Moreover, like other studies, the socio-demographic and lifestyle factors of participants involved in this study were linked to depressive symptoms, most notably lower SES, moderate/poor health status, trouble sleeping, chronic disease, and smoking. Our findings suggest the need for health education programs and interventions (e.g., depression screening) for people infected with COVID-19 in Bangladesh. The healthcare authority should be concerned about the individuals who have recovered from COVID-19 in terms of providing care to them and monitoring their health (including mental health) after their recovery from COVID-19. Virtual awareness programs should be considered with a view to mitigating self-medication practices without consultation. Individuals with persistent COVID-like symptoms should have consultations with healthcare professionals. Mental health counseling through e-health may be one approach for treating depression among people during times of spatial distancing. Our findings provide baseline information that may help guide healthcare provision and policy.

## Figures and Tables

**Figure 1 ijerph-18-01453-f001:**
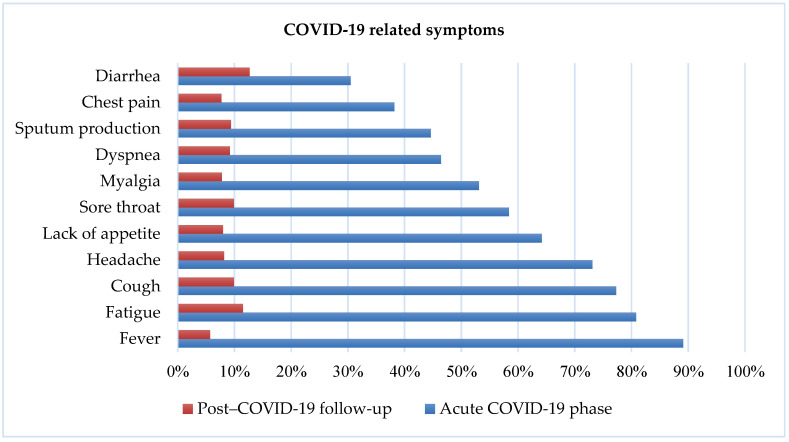
COVID-19-related symptoms during COVID-19 and post COVID-19 stage.

**Table 1 ijerph-18-01453-t001:** Characteristics of people who were infected by severe acute respiratory syndrome coronavirus 2 (SARS-CoV-2) (experienced positive coronavirus disease (COVID-19) testing) and regression analysis predicting depression.

Variables	Total		Depression					
			Negative		Positive ^†^		χ^2^	AOR (95% CI)
	n	(%)	n	(%)	n	(%)		
Age								-
Younger adults (18–39 years)	653	(65.2)	355	(54.4)	298	(45.6)	4.58	
Middle-aged/older adults (>40 years)	349	(34.8)	165	(47.3)	184	(52.7)		
Sex								
Male	580	(57.9)	191	(45.3)	231	(54.7)	12.86 **	
Female	422	(42.1)	329	(56.7)	251	(43.3)		
Marital status								
Unmarried	436	(43.5)	242	(45.5)	290	(54.5)	19.88 **	
Married	532	(53.1)	17	(50.0)	17	(50.0)		
Divorced/widowed	34	(3.4)	261	(59.9)	175	(40.1)		
Family type ^‡^								-
Nuclear	711	(71.0)	132	(45.4)	159	(54.6)	7.02	
Joint	291	(29.0)	388	(54.6)	323	(45.4)		
Socioeconomic status (SES)								
Lower	259	(25.8)	91	(35.1)	168	(64.9)	55.34 **	2.22 ** (1.51–3.26)
Middle	345	(34.4)	172	(49.9)	173	(50.1)		1.82 ** (1.3–2.55)
Upper	398	(39.7)	257	(64.6)	141	(35.4)		Reference
Health status								
Moderate/poor	532	(53.1)	338	(71.9)	132	(28.1)	142.11 **	3.17 ** (2.33–4.33)
Good	470	(46.9)	182	(34.2)	350	(65.8)		Reference
BMI								-
Normal/lean	478	(47.7)	31	(62.0)	19	(38.0)	5.21	
Underweight	50	(5.0)	230	(48.5)	244	(51.5)		
Overweight/obese	474	(47.3)	259	(54.2)	219	(45.8)		
Sleep disturbance								
Yes	523	(52.2)	216	(41.3)	307	(58.7)	49.20 **	1.97 ** (1.46–2.64)
No	479	(47.8)	304	(63.5)	175	(36.5)		Reference
Smoking								-
Yes	123	(12.3)	62	(50.4)	61	(49.6)	0.13	
No	879	(87.7)	458	(52.1)	421	(47.9)		
Physical exercise								
Yes	279	(27.8)	343	(47.4)	380	(52.6)	20.64 **	
No	723	(72.2)	177	(63.4)	102	(36.6)		
Diabetes								
Yes	211	(21.1)	76	(36.0)	135	(64.0)	26.99 **	
No	791	(78.9)	444	(56.1)	347	(43.9)		
Hypertension/high blood pressure								
Yes	249	(24.9)	101	(40.6)	148	(59.4)	17.05 **	
No	753	(75.1)	419	(55.6)	334	(44.4)		
Heart disease								
Yes	82	(8.2)	28	(34.1)	54	(65.9)	11.27 *	
No	920	(91.8)	492	(53.5)	428	(46.5)		
Cancer								
Yes	24	(2.4)	5	(20.8)	19	(79.2)	9.50 *	
No	978	(97.6)	515	(52.7)	463	(47.3)		
Kidney problems								-
Yes	59	(5.9)	29	(49.2)	30	(50.8)	0.19	
No	943	(94.1)	491	(52.1)	452	(47.9)		
Asthma/respiratory problems								
Yes	255	(25.4)	98	(38.4)	157	(61.6)	24.84 **	1.73 * (1.23–2.43)
No	747	(74.6)	422	(56.5)	325	(43.5)		Reference
Fear of re-infection								
Yes	629	(62.8)	254	(40.4)	375	(59.6)	89.74 **	2.16 ** (1.58–2.97)
No	373	(37.2)	266	(71.3)	107	(28.7)		Reference
Self-medication								
Yes	242	(24.2)	104	(43.0)	138	(57.0)	10.17 *	
No	760	(75.8)	416	(54.7)	344	(45.3)		
Persistent symptoms								
Yes	200	(20.0)	59	(29.5)	141	(70.5)	50.21 **	
No	802	(80.0)	461	(57.5)	341	(42.5)		
Hospitalization								
Yes	208	(20.8)	78	(37.5)	130	(62.5)	21.79 **	
No	794	(79.2)	442	(55.7)	352	(44.3)		

AOR: adjusted odds ratio; CI: confidence interval; BMI: body mass index. ^†^ Positive depression indicates moderate to severe depression (Patient Health Questionnaire (PHQ-9) ≥ 10); ^‡^ Nuclear (two parents and their children)/ joint (family unit with more than two parents, extended family); * *p* < 0.003, ** *p* < 0.001.

**Table 2 ijerph-18-01453-t002:** Bivariate analyses using self-medication, persistent symptoms, and hospitalization as dependent variables.

Variables	Hospitalization			Self-Medication			Persistent Symptoms		
	Yes	No	χ^2^	Yes	No	χ^2^	Yes	No	χ^2^
	*n* (%)	*n* (%)		*n* (%)	*n* (%)		*n* (%)	*n* (%)	
Age									
Younger adults	77 (11.8)	576 (88.2)	91.64 **	131 (20.1)	522 (79.9)	17.12 **	103 (15.8)	550 (84.2)	20.57 **
Middle-aged/older adults	131 (37.5)	218 (62.5)		111 (31.8)	238 (68.2)		97 (27.8)	252 (72.2)	
Sex									
Male	99 (17.1)	481 (82.9)	11.40 *	117 (20.2)	463 (79.8)	11.90 *	91 (15.7)	489 (84.3)	15.72 **
Female	109 (25.8)	313 (74.2)		125 (29.6)	297 (70.4)		109 (25.8)	313 (74.2)	
Marital status									
Unmarried	52 (11.9)	384 (88.1)	38.76 **	98 (22.5)	338 (77.5)	3.46	77 (17.7)	359 (82.3)	2.58
Married	150 (28.2)	382 (71.8)		139 (26.1)	393 (73.9)		116 (21.8)	416 (78.2)	
Divorced/widowed	6 (17.6)	28 (82.4)		5 (14.7)	29 (85.3)		7 (20.6)	27 (79.4)	
Family type ^‡^									
Nuclear	118 (16.6)	593 (83.4)	25.78 **	163 (22.9)	548 (77.1)	2.01	121 (17)	590 (83)	13.26 **
Joint	90 (30.9)	201 (69.1)		79 (27.1)	212 (72.9)		79 (27.1)	212 (72.9)	
Socioeconomic status (SES)									
Lower	96 (37.1)	163 (62.9)	56.76 **	101 (39)	158 (61)	42.92 **	92 (35.5)	167 (64.5)	57.94 **
Middle	49 (14.2)	296 (85.8)		71 (20.6)	274 (79.4)		38 (11)	307 (89)	
Upper	63 (15.8)	335 (84.2)		70 (17.6)	328 (82.4)		70 (17.6)	328 (82.4)	
Health status									
Good	61 (13)	409 (87)	32.57 **	89 (18.9)	381 (81.1)	13.15 **	53 (11.3)	417 (88.7)	41.78 **
Moderate/poor	147 (27.6)	385 (72.4)		153 (28.8)	379 (71.2)		147 (27.6)	385 (72.4)	
BMI									
Normal/lean	73 (15.3)	405 (84.7)	27.33 **	107 (22.4)	371 (77.6)	2.56	75 (15.7)	403 (84.3)	15.29 **
Underweight	4 (8)	46 (92)		10 (20)	40 (80)		6 (12)	44 (88)	
Overweight/obese	131 (27.6)	343 (72.4)		125 (26.4)	349 (73.6)		119 (25.1)	355 (74.9)	
Sleep disturbance									
Yes	136 (26)	387 (74)	18.30 **	127 (24.3)	396 (75.7)	0.01	143 (27.3)	380 (72.7)	37.31 **
No	72 (15)	407 (85)		115 (24)	364 (76)		57 (11.9)	422 (88.1)	
Smoking									
Yes	26 (21.1)	97 (78.9)	0.01	28 (22.8)	95 (77.2)	0.15	26 (21.1)	97 (78.9)	0.12
No	182 (20.7)	697 (79.3)		214 (24.3)	665 (75.7)		174 (19.8)	705 (80.2)	
Physical exercise									
Yes	58 (20.8)	221 (79.2)	<0.001	61 (21.9)	218 (78.1)	1.11	58 (20.8)	221 (79.2)	0.17
No	150 (20.7)	573 (79.3)		181 (25)	542 (75)		142 (19.6)	581 (80.4)	
Diabetes									
Yes	87 (41.2)	124 (58.8)	68.11 **	71 (33.6)	140 (66.4)	13.16 **	72 (34.1)	139 (65.9)	33.56 **
No	121 (15.3)	670 (84.7)		171 (21.6)	620 (78.4)		128 (16.2)	663 (83.8)	
Hypertension									
Yes	67 (26.9)	182 (73.1)	7.62	54 (21.7)	195 (78.3)	1.10	49 (19.7)	200 (80.3)	0.01
No	141 (18.7)	612 (81.3)		188 (25)	565 (75)		151 (20.1)	602 (79.9)	
Heart disease									
Yes	29 (35.4)	53 (64.6)	11.59 *	31 (37.8)	51 (62.2)	9.09	26 (31.7)	56 (68.3)	7.71
No	179 (19.5)	741 (80.5)		211 (22.9)	709 (77.1)		174 (18.9)	746 (81.1)	
Cancer									
Yes	14 (58.3)	10 (41.7)	21.11 **	15 (62.5)	9 (37.5)	19.74 **	13 (54.2)	11 (45.8)	18.01 **
No	194 (19.8)	784 (80.2)		227 (23.2)	751 (76.8)		187 (19.1)	791 (80.9)	
Kidney problems									
Yes	23 (39)	36 (61)	12.656 **	26 (44.1)	33 (55.9)	13.57 **	20 (33.9)	39 (66.1)	7.62
No	185 (19.6)	758 (80.4)		216 (22.9)	727 (77.1)		180 (19.1)	763 (80.9)	
Asthma/respiratory problems									
Yes	59 (23.1)	196 (76.9)	1.18	62 (24.3)	193 (75.7)	0.01	53 (20.8)	202 (79.2)	0.15
No	149 (19.9)	598 (80.1)		180 (24.1)	567 (75.9)		147 (19.7)	600 (80.3)	
Fear of re-infection									
Yes	165 (26.2)	464 (73.8)	30.78 **	165 (26.2)	464 (73.8)	3.99	177 (28.1)	452 (71.9)	70.77 **
No	43 (11.5)	330 (88.5)		77 (20.6)	296 (79.4)		23 (6.2)	350 (93.8)	
Depression									
Negative	78 (15)	442 (85)	21.79 **	104 (20)	416 (80)	10.17 *	59 (11.3)	461 (88.7)	50.21 **
Positive	130 (27)	352 (73)		138 (28.6)	344 (71.4)		141 (29.3)	341 (70.7)	
Hospitalization									
Yes	208 (100)	0 (0)	―	76 (36.5)	132 (63.5)	21.99 **	106 (51)	102 (49)	157.91 **
No	0 (0)	794 (100)		166 (20.9)	628 (79.1)		94 (11.8)	700 (88.2)	
Self-medication									
Yes	76 (31.4)	166 (68.6)	21.99 **	242 (100)	0 (0)	―	85 (35.1)	157 (64.9)	45.92 **
No	132 (17.4)	628 (82.6)		0 (0)	760 (100)		115 (15.1)	645 (84.9)	
Persistent symptoms									
Yes	106 (53)	94 (47)	157.91 **	85 (42.5)	115 (57.5)	45.92 **	200 (100)	0 (0)	―
No	102 (12.7)	700 (87.3)		157 (19.6)	645 (80.4)		0 (0)	802 (100)	

^‡^ Nuclear (two parents and their children)/ joint (family unit with more than two parents, extended family); * *p* < 0.003, ** *p* < 0.001.

**Table 3 ijerph-18-01453-t003:** Multivariate regression analyses for hospitalization, self-medication, and persistent symptoms showing significant adjusted odds ratios.

Variables	Hospitalization	Self-Medication	Persistent Symptoms
	AOR (95% CI)	AOR (95% CI)	AOR (95% CI)
Age			
Middle-aged/older adults	2.97 ** (1.85–4.77)	-	-
Younger adults	Reference		
Socioeconomic status (SES)			
Lower	2.04 * (1.28–3.23)	2.31 ** (1.56–3.41)	0.93 (0.59–1.48)
Middle	0.97 (0.62–1.53)	1.28 (0.88–1.88)	0.45 * (0.28–0.72)
Upper	Reference	Reference	Reference
Fear of re-infection			
Yes	-	-	3.32 ** (2.01–5.47)
No			Reference
Hospitalization			
Yes	-	-	4.89 ** (3.22–7.44)
No			Reference
Self-medication			
Yes	-	-	2.23 ** (1.47–3.37)
No			Reference
Persistent symptoms			
Yes	4.9 ** (3.19–7.51)	2.14 ** (1.44–3.18)	-
No	Reference	Reference	

AOR: adjusted odds ratio; CI: confidence interval; ** p* < 0.003, ** *p* < 0.001.

## Data Availability

The data presented in this study are available in (Appendix A here).

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
