# Peer review of "Treatment, Persistent Symptoms, and Depression in People Infected with COVID-19 in Bangladesh"

_ijerph, 2021, doi:10.3390/ijerph18041453_

Round 1

Reviewer 1 Report

In this cross-sectional online survey on a total of 1002 participants, the authors aimed to examine the data pertaining to initial treatment and persistent COVID-19 symptoms and whether COVID-19 led to changes in depressive symptoms in people who had been infected with COVID-19 in Bangladesh.

The article is well written. It is logically divided into sections. Statistical analyses are appropriate and well performed. The methodology is well explained. The English language is well used, with some minor mistakes. I would advise the authors to check the articles, I feel they missed articles in a few places. The flow of the article was fine, I had no problems reading it (except for the very large tables).
I ran the article through the PlagScan software, and it seems there are no issues (the sofware reported a high level of similarity, 13.8%, but most of the similarities are related to common phrasing and references).

Here are some more specific complaints:

Methodology:
How was informed consent gathered? Were measures done to assure the authenticity of the data? This is mentioned in the Ethics section, but I feel it should be elaborated further, as online surveys are often considered to be a "gray area" related to informed consent.
Results:
I feel tables 1 and 2 are too large. I think it might be able to reduce the size of tables 1 and 4 by putting the variable titles before the answers, in a separate column, this way, the table would be wider, but shorter, and possibly, to increase the font.
Table 2 seems overwhelming. It is simply too large, and doesn't fit on a single page. I would advise to divide it into 2 tables, by some criterion of the authers choice. Second option, No and (%) could be fit in a single column, thus making it a bit narrower. That way, the variables might be put in a separate column, making the table shorter.
Is there a reason the authors chose *p<0.003 as a limit, and not the standard 0.05, 0.005 and 0.001? If there is a specific reason, or newer standard I am not aware of, it might be useful to mention it in the statistics section, or in a reply to this peer review.
Discussion:
I feel the discussion section is much too short when considering the sheer amount of data the authors provided. Specifically the data pertaining to the sociodemografics. This is related also to my previous comments: much space was given to the sociodemografics in the results section, many of the results aren't expanded upon.
How do your results compare to other research?
How often do people self-medicate in other countries and in previous research in your country?
I would like to see comparisons of the prevalence of depression prior to COVID, possibly in Bangladesh, with your results.
Not much space was given to the fact that a large proportion of your participants were younger adults. In what way can that affect your results?
I feel as though the discussion section lacks behind the rest of the article, which I consider to be very well written.

Conclusion:
I feel the Conclusion section is too vague. I would like to see concrete conclusions that stem from your research. What are the most interesting results? Was something surprising, or was everything in line with your expectations?
Line 321-322: "this information can be used to inform prevention and management strategies for public health policymakers for people in Bangladesh" - In what way? What are the practical implications?

Line 22: A space is missing between COVID-19 and symptoms.
Line 22-23: Saying " whether COVID-19 led to changes in the mental status of people" is an overstatement, since the authors analyzed only depressive symtoms.
Line 43: "novel coronavirus" is written with a smaller font.
Line 44: I would prefer it said: the number of cases continue to increase.
(Optional) Line 54: It would be useful if the authors could give newer data (if available), since this data is 2 months old.
Line 58: "Recovery is often not complete" isn't completely clear. It would be better to write total, instead of complete.
Line 63: "experiencing one or two persistent symptoms". After what? The the onset, the cessation of symptoms, or after a negative test result? This shoul be specified.
Line 76: It would be clearer to say: "including higher mortality", or "higher mortality rates."
Line 94-95: Something is missing in this sentence: "The questionnaire was translated using the guidelines of back translation [24] as previously in Bangladesh [25–27]. "
Line 113: Response rate is a proportion of participants willing to be included in a study, of the total number of participants asked. In this line, I can pressume you meant: "Our sample size"
Line 285: A space is missing between "their" and "COVID"

Reviewer 2 Report

Thank you for the opportunity to review this manuscript, which considers some interesting, applied issues. This study appears to be novel, but as submitted needs considerable work on the presentation. The authors showed an interesting point about the “Treatment, persistent symptoms and depression in people infected with COVID-19 in Bangladesh”, unfortunately, there are few points to overcome.

Line 23: these!!! Please to clarify

Line 105 please to include the exclusion criteria

Considering the complex nature of this investigation, I suggest a mathematic model to explain this results https://pubmed.ncbi.nlm.nih.gov/30890960/

To include a multivariate analysis in Statistical section

I suggest to use the same style on significant values “0.00 or .00” please check the main document

Reviewer 3 Report

Dear Authors,

thank you for possibility of reading your paper.

This paper examines the issue of initial treatment, persistent COVID-19 symptoms and mental health in area of depression among 1002 people who have already experienced COVID-19 infection in Bangladesh. The study tackles very important issue of post-COVID symptoms and mental health. I believe that this will be a crucial topic requiring in-depth exploration and research in the nearest future. Therefore, this paper relates to an urgent  problem of post-COVID well-being.

     Abstract:

  1. line 23: In the Background Authors underline “changes” in the mental status due to the COVID-19, yet this paper does not answer this issue. As Authors showed in the Limitation part, this is not a longitudinal study, therefore in does not refer to any changes. Even though the participants have assessed their COVID related symptoms during infection and after infection, there is no information about their levels of depression during or before COVID-19 infection.
  2. line 25: I don’t understand why Authors call participants “patients”, as only 21% of them were hospitalized. Yest in the Introduction, Authors underline that the study was conducted within a group of hospitalized people (line: 87). Also, in the Participants and procedure part, there is only a superficial description of a snowballing It should be elaborated.
  3. Authors claim to conduct hierarchical regression analysis (line: 34), yet, in the body manuscript, there is other kind of regression analysis used.

Introduction

  1. The medical situation in Bangladesh was well explored, yet there are major shortcomings when it comes to depression. Firstly, there is a lack of a theory behind the presented model of depression, as well as there is a lack of a literature review in this topic. Secondly, the depression predictors are not well explained in the introduction, therefore, there is no justification for hypothesis, that by the way, were not clarified in this paper as well. Without a theoretical justification for regression model for depression, the scientific soundness of this paper lowers down.

Materials and Methods

  1. Point 2.1. participants and procedure: in line 92 the Authors inform, that participants had experienced COVID-19 over a month period (and reported positive tests for COVID-19: line 183), between September and October, 2020. Would you be more precise, to what extend “over a month period” refers to. Considering, that this is the most important issue this research, it should be clarified. When exactly the study was conducted? Also, there could be a difference between respondents who were diagnosed a one-moth period or a three-month period before participating in the study. Elaborated this part, please.

  1. Point 2.4.4. Lines 160-161: Could you elaborate about exact item(s)? referring to worry about re-infection?

  1. My main concern is the lack of monitoring the history of medical treatment before COVID-19. The analyzed symptoms of COVID-19 infection are specific for COVID-19 only. It may fit to the image of any other disease. The participants with decreased immunity system or other diseases could presented those symptoms after COVID-19 infection but not due to it. the same issue refers to depression symptoms. We have no knowledge whether the participants had diagnosed depression before COVID-19 infection. This a serious flaw of the study design.

Results

  1. The description of depression levels is to simplistic. All five levels should be described.

Round 2

Reviewer 3 Report

Dear Authors,

Thank you for adjusting your paper.

All of my remarks have been included except for: „Firstly, there is a lack of a theory behind the presented model of depression, as well as there is a lack of a literature review in this topic”.  Please, refer to the theoretical basis of depression. Although depression is the topic of this paper there is no definition of it.

I accept all other changes in the paper.
